# Meta-Analysis and MaxEnt Model Prediction of the Distribution of *Phenacoccus solenopsis* Tinsley in China under the Context of Climate Change

**DOI:** 10.3390/insects15090675

**Published:** 2024-09-06

**Authors:** Zhiqian Liu, Yaqin Peng, Danping Xu, Zhihang Zhuo

**Affiliations:** College of Life Science, China West Normal University, Nanchong 637002, China; qnhtvxhp319123@foxmail.com (Z.L.); pengyaqin2023@foxmail.com (Y.P.); xudanping@cwnu.edu.cn (D.X.)

**Keywords:** *Phenacoccus solenopsis Tinsley*, meta-analysis, MaxEnt, environmental indicators, potential distribution

## Abstract

**Simple Summary:**

To predict the future distribution of *Phenacoccus solenopsis Tinsley*, this study combines MaxEnt modeling and meta-analysis. The MaxEnt model effectively predicts species distribution by analyzing environmental variables and occurrence data, providing accurate predictions of potential suitable habitats even with limited data. Meanwhile, the meta-analysis aggregates data from multiple experimental studies to assess *P. solenopsis*’s response to temperature changes. By combining these methods, we can enhance prediction reliability and delve into the potential impacts of climate change on pest spread. This integrated approach offers detailed and reliable forecasts of *P. solenopsis*’s future distribution, providing valuable insights for effective pest management and agricultural planning.

**Abstract:**

*Phenacoccus solenopsis Tinsley* is a pest that poses a significant threat to agricultural crops, especially cotton, and is now widely distributed across many regions worldwide. In this study, we performed a meta-analysis on the collected experimental data and found that within the suitable temperature range, the survival rate of *P. solenopsis* increases with rising temperatures, indicating that climate plays a decisive role in its distribution. Using the MaxEnt model this study predicted that under three future climate scenarios (SSP1–2.6, SSP3–7.0, and SSP5–8.5), the distribution of *P. solenopsis* will expand and move towards higher latitudes. Climate change is the primary factor influencing changes in pest distribution. We conducted a meta-analysis of *P. solenopsis*, including seven independent studies covering 221 observation results, and examined the impact of temperature ranging from 18 °C to 39 °C on the developmental cycle of *P. solenopsis*. As the temperature rises, the development cycle of *P. solenopsis* gradually decreases. Additionally, by combining the MaxEnt model, we predicted the current and potential future distribution range of *P. solenopsis*. The results show that under future climate warming, the distribution area of *P. solenopsis* in China will expand. This research provides a theoretical basis for early monitoring and control of this pest’s occurrence and spread. Therefore, the predictive results of this study will provide important information for managers in monitoring *P. solenopsis* and help them formulate relevant control strategies.

## 1. Introduction

*Phenacoccus solenopsis Tinsley* is an invasive pest that threatens the safe production of economic crops such as agricultural crops, horticultural plants, fruit trees, and vegetables in recent years. *P. solenopsis* belongs to the family Pseudococcidae, superfamily Coccoidea, suborder Sternorrhyncha, and order Hemiptera. Originating from North America, *P. solenopsis* was initially discovered in a park in New Mexico, USA [1], with host plants including *Boerhavia spicata* and *Kallstroemia brachystylis*. Subsequently, it has spread gradually to other states in the United States, South America, Africa, Asia, Europe, and Australia [2]. Its presence was first recorded in Guangzhou, Guangdong Province, China, in 2008 [3]. Additionally, *P. solenopsis* primarily infests plants such as eggplant, pumpkin, tomato, sunflower, euphorbia, castor bean, pigeon pea, okra, and pigweed, posing a significant threat to major economic crops like cotton [4]. It mainly damages the tender parts of host plants, including young branches, leaves, flower buds, and petioles. When the population density is high, it can also parasitize older branches and main stems. The female and nymphs feed on plant sap, causing damage. Affected plants grow slowly or stop growing, leading to wilting and dehydration. Its honeydew can induce sooty mold, which hinders photosynthesis and may cause leaf detachment, and in severe cases, plant death [5]. Therefore, predicting the suitable distribution areas of *P. solenopsis* under future climate change conditions is especially crucial.

Species Distribution Models (SDMs), also known as ecological niche models, are a method used to estimate species distribution in geographic space based on the actual distribution of species. They are commonly employed to predict the relationship between species distribution and the environment. In insect biogeography and regional ecology, SDMs are widely used to map the potential distribution of insects under global climate change [6,7]. The application of species distribution models helps monitor changes in species migration, thereby aiding in the formulation of better management strategies.

The MaxEnt model, proposed by Phillips in 2004 [8,9], is a type of SDM that utilizes environmental factors such as maximum temperature, minimum temperature, relative humidity, and precipitation to infer the potential geographic distribution of species [10,11,12]. The MaxEnt model demonstrates excellent predictive accuracy, particularly in cases where species distribution data are limited. It achieves this by quantitatively describing the potential habitat of species through the selection of key ecological environmental factors, thus enabling the simulation of species habitat distribution. Consequently, the MaxEnt model has been widely applied in predicting the potential distribution of invasive species and species under climate change [13,14,15].

## 2. Materials and Methods

### 2.1. Meta-Analysis

#### 2.1.1. Data Collection and Extraction

In this study, we conducted a targeted literature search using the China National Knowledge Infrastructure (CNKI) database (http://www.cnki.net/, accessed on 27 March 2024) and Web of Science, with “temperature” and “*Phenacoccus solenopsis Tinsley*” as the main keywords. During the search process, two specific criteria were applied to limit the search results: (1) experiments must be related to *P. solenopsis*, and (2) one of the experimental treatments must involve temperature changes. Ultimately, the search identified 16 relevant articles. Based on the following selection criteria, the articles were screened: (1) the temperature range set in the study includes 18 °C, (2) the results of the study include the developmental cycle of adult/eggs/pupae/larvae stages, and (3) the sample size, mean, standard deviation, or standard error of the data provided. After screening, qualified independent studies were identified, collecting a total of 221 datasets. The collected data not only include temperature but also information on humidity, photoperiod, and host type from different studies. These data will be used to interpret the analysis results. During data processing, the experimental data collected were analyzed with the lowest temperature (18 °C) as the control group and other temperature categories as treatment groups, and various indicators were extracted. The purpose of focusing on 18 °C is to better capture the changes in capture volume and the growth rate of *P. solenopsis* under increasing temperature conditions. We ensured that the collected datasets contain the corresponding standard deviation (SD). If the literature provides standard error (SE), it needs to be converted to standard deviation (SD) using a formula, typically by dividing the standard error by the square root of the sample size. The calculation method is shown in Formula (1).
(1)$$SD=SE\sqrt{n}$$

In Formula (1), “*n*” represents the sample size.

#### 2.1.2. Data Analysis

In this study, a meta-analysis was conducted on the collected and filtered data to evaluate whether elevated temperatures would reduce the developmental period of *P. solenopsis*. The effect size for each observation was computed as the natural logarithm transformed (*ln*) response ratio (*RR*), as shown in Formula (2):(2)$$RR=ln(\frac{X_{t}}{X_{c}})=ln(X_{t})-ln(X_{c})$$

In Formula (2), *X_t_* and *X_c_* represent the mean values of the developmental periods of *P. solenopsis* in the experimental group and control group, respectively. The variance (*v*) (Formula (3)) of the RR is calculated as follows (Formula (4)):(3)$$v=\frac{s_{t}^{2}}{n_{t}{X_{t}}^{2}}+\frac{s_{c}^{2}}{n_{c}X_{c}^{2}}$$

In Formula (3), n*_t_* and n*_c_* represent the sample sizes of the experimental and control groups, respectively. Similarly, *s_t_* and *s_c_* represent their respective standard deviations (SD). When SD or SE isn’t provided in the study, 1/10 of the mean value is specified as the standard deviation. The weighted average response (*RR*^+^) is obtained by weighting the response ratio of each independent study.
(4)$$RR^{+}=\frac{\sum_{i=1}^{m}w_{i}(RR_{i})}{\sum_{i=1}^{m}w_{i}}$$

*m* is the number of comparisons in the group, *w_i_* is the weighting factor for the i-th experiment in the group, and *w_i_* is calculated as follows (Formula (5)):(5)$$w_{i}=\frac{1}{v_{i}}$$

The standard error (*RR*^+^) and 95% confidence interval (*CI*) are calculated as Formulas (6) and (7), respectively.
(6)$$s(RR^{+})=\sqrt{\frac{1}{\sum_{i=1}^{m}w_{i}}}$$
(7)$$95\%CI=lnRR^{+}\pm 1.96s(lnRR^{+})$$

For whether to introduce explanatory variables into the calculation results, *Q_t_* is used to determine the following (Formula (8)):(8)$$Q_{t}=Q_{m}+Q_{e}$$

In the formula, *Q_m_* represents the heterogeneity caused by a known factor, where a higher value indicates a greater influence of the explanatory variable on the effect size. On the other hand, *Q_e_* represents the unexplained residual heterogeneity. When the total heterogeneity *Q_t_* is high, it indicates that data points deviate significantly from the mean, possibly due to other factors causing such deviations. If the data are homogeneous, *Q_t_* should follow a chi-square distribution with k−1 degrees of freedom, and in this case, there is no need to introduce explanatory variables.

In this study, the “rma.mv” function from the R package “metafor” in version 4.3 was employed for the following steps. Firstly, the relative risk (*RR*^+^) was calculated using a random-effects model, and the variance between cases was estimated using restricted maximum likelihood (REML). Explanatory variables were introduced based on the value of *I*^2^. Subsequently, a random-effects model was used to compute the overall average effect size for all treatment group temperatures. Finally, all statistical tests were conducted, including the analysis of the average effect size, 95% confidence intervals (*CI*), *Q_t_*, and *I*^2^.

The heterogeneity statistic is a test of the weighted sum of squares against a chi-square distribution with *k* − 1 degrees of freedom. When the 95% confidence interval of the effect size includes 0, it indicates that the effect size of the experimental group is equal to that of the control group, and the impact of both on the study subject is comparable (*p* > 0.05). When the values of the 95% confidence interval are all greater than 0, it indicates that the effect size of the experimental group is greater than that of the control group (*p* < 0.05). Conversely, when the values of the 95% confidence interval are all less than 0, it indicates that the effect size of the experimental group is less than that of the control group (*p* < 0.05). Based on the significance of the relationship between the effect size and zero, explanatory variables were included in the meta-analysis, including (1) developmental stage (adult, first instar nymph, second instar nymph, third instar nymph, larvae, egg); (2) humidity; (3) photoperiod; (4) host plant. Additionally, temperature data were treated as a continuous variable to determine its effect on the average effect size. In the meta-analysis, overall heterogeneity is divided into variance explained by categorical factors (between-group heterogeneity) and residual variance (within-group heterogeneity), and their significance can be determined through *k* − 1 testing. In meta-regression analysis, explanatory variables (such as developmental stage, humidity, photoperiod, etc.) are treated as independent variables, and the effect size is treated as the dependent variable to examine the impact of these factors on the variation in effect size.

To test for potential publication bias, a funnel plot was used to examine the relationship between effect size and sample size. The significance of the *p*-value can indicate whether publication bias exists in the study. To further assess the impact of publication bias on the results, the trim-and-fill method was employed. If the statistically significant results remain unchanged after correction, it can be concluded that the results are stable and not influenced by publication bias.

### 2.2. MaxEnt Prediction and Analysis

#### 2.2.1. Species Occurrence Data

Through extensive searches of online databases and referencing the Global Biodiversity Information Facility (GBIF, https://www.gbif.org/, accessed on 27 March 2024), distribution records of *P. solenopsis* were collected. When precise geographic coordinates were lacking in species distribution records, Google Maps (http://ditu.google.cn/, accessed on 27 March 2024) was utilized to determine longitude and latitude. Ultimately, 460 distribution points of *P. solenopsis* were obtained. To reduce the impact of data duplication and redundancy on the results, ENMTools 1.4 was utilized to filter distribution points. With a spatial resolution set to 2.5 arc-minutes (approximately 4.5 km), it was ensured that only one distribution point existed per grid cell [16,17]. Consequently, we obtained 150 distribution points of *P. solenopsis*.

#### 2.2.2. Environmental Variables

The 20 environmental variables used in this study were sourced from the WorldClim global climate database (http://www.worldclim.org, accessed on 27 March 2024), with a spatial resolution of 2.5 min. These environmental variables include one topographic variable (elevation) and climate data for five periods: current (1970–2000) and future projections for the 2030s (2021–2040), 2050s (2041–2060), 2070s (2061–2080), and 2090s (2081–2100). For future climate data, the ACCESS-CM2 model from global climate models (GCM) was utilized, with 19 bioclimatic variables selected for three shared socioeconomic pathways (SSP126, SSP370, and SSP585) (Table 1).

We imported the filtered distribution points into ArcGIS 10.8 and utilized the “Toolbox/Spatial Analyst Tools/Extraction/Sample” tool to import 22 environmental variables and conduct point interpolation sampling for the distribution points. Similarly, the initial MaxEnt prediction model was imported using the same method, and point interpolation sampling was conducted for the distribution points. In SPSS, multicollinearity analysis was performed using the variance inflation factor (VIF) and Spearman correlation analysis for the point interpolation data. We preliminarily screened out environmental factors with VIF values less than 100 and correlations less than 0.8. The VIF, also known as the reciprocal of tolerance, indicates multicollinearity between factors. When VIF < 10, there is no multicollinearity between factors; when 10 < VIF < 100, there is multicollinearity between factors; when VIF > 100, there is multicollinearity between factors [18,19]. Environmental variables are important parameters for constructing ecological niche models [20], and redundant environmental variables can lead to overfitting of results, thereby reducing model accuracy. In this study, ENMTools.pl software v. 1. 4. 4 was used to perform correlation analysis on these 22 environmental variables. When the Pearson correlation coefficient is |>0.8|, it is defined as a highly correlated variable [21]. The contribution rates of both variables in the initial model were compared, and the variable with higher contribution rates and easily analyzable factors was retained [22] (Table 2).

#### 2.2.3. Optimization of Model Parameters

Based on statistical significance (partial receiver operating characteristic (ROC) with 500 iterations), predictive ability (omission rate (OR)), and model complexity (AICc), model parameters were optimized using the kuenm_ceval function [23]. According to the “OR_AICc” criterion, significant candidate models with omission rates below the threshold (e.g., ≤0.05) and the lowest ΔAICc values (≤2) are considered to have the best model parameters [24].The parameters of the MaxEnt model include feature combinations (FC), regularization multiplier (RM), maximum background points (BC), etc. [25]. Currently, MaxEnt has five features: linear (L), quadratic (Q), hinge (H), product (P), and threshold (T) [26]. In the default setting, the value of RM is 1, and the selection and use of specific feature combinations depend on the number of species distribution points. Generally, linear features are always used, quadratic features are used when the number of species distribution points exceeds 10, hinge features are used when it exceeds 15, and threshold and product features are used only when the distribution points exceed 80 [27]. However, studies have shown that the default settings of MaxEnt may not be suitable for predicting the distribution of all species, which may result in overfitting and difficulty in explaining the final prediction results [28].

To optimize model parameters, Muscarella developed an R package called ENMeval. This program analyzes the complexity of models under various parameter conditions by changing the regularization multiplier and feature combination parameters and selects the model parameters with the lowest complexity for modeling [29]. In this study, the ENMeval package in R 3.6.3 was used to optimize the MaxEnt model. First, *P. solenopsis* was divided into four equal parts using a partitioning method, with three parts used for training and the remaining part for testing [30]. Then, in R 3.6.3, we used the kuenm package to compare various combinations of the two most critical parameters (feature class and regularization multiplier) to select the optimal combination [31]. MaxEnt contains a total of 31 feature combinations for the five features, with regularization multiplier values ranging from 0.1 to 4 at intervals of 0.1. A total of 1240 candidate models were evaluated, including all combinations of 40 regularization multiplier settings, 31 feature class combinations, and a set of 11 environmental variables.

#### 2.2.4. Shift of Suitable Habitat Distribution Center

In this study, the SDM Toolbox (http://www.sdmtoolbox.org/downloads, accessed on 27 March 2024), a Python-based GIS toolkit, was utilized to compute and compare the suitable habitat regions for *P. solenopsis* currently and in the future [32]. With the SDM Toolbox, we calculated the centroids of the suitable habitat zones for *P. solenopsis* and, under the same conditions, computed the centroids of the suitable habitat zones for *P. solenopsis* under future conditions [33]. Subsequently, the magnitude and direction of the centroid shift from the current to the future were plotted by us, facilitating further analysis of the research findings.

#### 2.2.5. Model Evaluation

The area under the curve (AUC) of the receiver operating characteristic (ROC) curve is a common method used to evaluate the accuracy of MaxEnt models. The AUC value provides a measure to assess the model’s ability to classify positive and negative samples correctly across different thresholds. Typically, the AUC value ranges from 0 to 1, where AUC < 0.5 indicates random prediction, 0.5 ≤ AUC < 0.7 indicates poor model performance, 0.7 ≤ AUC ≤ 0.9 indicates moderate performance, and AUC > 0.9 indicates high performance [34].

## 3. Results

### 3.1. Survival Response to Temperature

#### 3.1.1. Cumulative Effect Size

Calculating cumulative effect size is a method to determine whether the overall effect of treatment is significant. This process involves two model selections: fixed-effect model and random-effects model. The fixed-effect model considers only the variation of effect sizes within cases, while the random-effects model considers both within-case and between-case variations. The results (Appendix A) indicate that the cumulative effect sizes calculated by both models are less than zero and fall within the confidence intervals. This suggests that in the warming experiments conducted at temperatures ranging from 18 °C to 39 °C, the developmental period of *P. solenopsis* significantly decreases as the temperature rises.

Variations in temperature were observed to significantly affect the developmental period of *P. Tinsley* (Figure 1A). As temperatures increased, the developmental period of *P. Tinsley* gradually shortened, with varying degrees of influence observed at different temperature levels (Figure 2A). The results indicate that within the temperature range of 20–35 °C, the developmental period of *P. Tinsley* decreased progressively with rising temperatures, reaching a peak in the shortening trend between 30–35 °C (Figure 1B). At 32 °C, the developmental period of *P. Tinsley* was the shortest. These findings suggest that under conditions of temperature variation, warming notably impacts the shortened developmental period of *P. Tinsley*.

#### 3.1.2. Introduction of Explanatory Variables

An analysis of variables other than temperature was conducted to explain the differences among different environmental conditions. The results indicated that under different humidity and photoperiod conditions, there were significant differences in the impact of temperature increase on the developmental period of *P. Tinsley* (Figure 2B). Additionally, under different host plant conditions, the impact of temperature increase on *P. Tinsley* also varied significantly (Appendix A).

#### 3.1.3. Model Diagnostics

In this study, funnel plots and Egger’s regression test were employed to assess publication bias within the dataset. The results revealed significant asymmetry (*p* < 0.05), suggesting the presence of publication bias. To address this issue, the trim-and-fill method was utilized to impute hypothetical study points. However, the significance of the adjusted results remained unchanged, indicating that publication bias did not affect the model’s output. These findings are crucial for understanding the reliability of the dataset and the potential impact of publication bias on the results. Additionally, to ensure the accuracy and credibility of the results, a Q-Q plot was employed to validate the reliability of the findings (Appendix A).

### 3.2. Predicting the Distribution of P. solenopsis in China

#### 3.2.1. Prediction Accuracy Evaluation of MaxEnt Model

Among the 1240 candidate models, all exhibited statistical significance. By running the kuenm package in R 3.6.3, only one model that met both OR and AICc criteria was obtained. Therefore, in setting up the MaxEnt model for *P. solenopsis*, we selected model M_1.2_F_pt_Set_1 (regularization multiplier = 1.2, feature combination = P and T). Based on 460 current distribution records and 9 environmental variables, the potential geographic distribution of *P. solenopsis* in China was simulated using the MaxEnt software v. 3. 4. 4. The training AUC value was 0.980 (Figure 3), indicating an excellent level of performance. This suggests that the MaxEnt model’s predictive results are accurate and reliable, with high predictive ability.

#### 3.2.2. Main Environmental Factors Affecting Distribution

The environmental factors influencing the distribution range of *P. solenopsis*, as determined by the MaxEnt model, are as follows: the contribution rates of the most significant factors are as follows: precipitation of the warmest quarter (Bio18) at 40.6%; mean temperature of the wettest quarter (Bio08) at 21%; isothermality (Bio03) at 17.9%; precipitation of the wettest month (Bio13) at 5.5%; elevation at 4.5%; precipitation of the driest month (Bio14) at 3.7%; mean temperature of the driest quarter (Bio09) at 3.6%; precipitation seasonality (coefficient of variation) (Bio15) at 2.8%; and slope at 0.3% (Table 3). Blade-cutting experiments revealed that using only a single ecological factor, the four most significant environmental variables affecting the distribution were precipitation of the wettest quarter (Bio8), mean temperature of the driest quarter (Bio09), precipitation of the wettest month (Bio13), and precipitation of the warmest quarter (Bio18) (Figure 4).

Based on the comprehensive analysis results, we can determine that the primary environmental factors influencing the distribution of *P. solenopsis* are temperature and precipitation. Specifically, the mean temperature of the wettest quarter, precipitation of the warmest quarter, mean temperature of the driest quarter, and precipitation of the wettest month play critical roles in the distribution of *P. solenopsis*. According to the response curves of *P. solenopsis*’s main environmental factors (Figure 5), we can draw the following conclusions: when the probability of *P. solenopsis* distribution is ≥0.66, indicating a high suitability level, the mean temperature of the wettest quarter ranges from 27.26 to 28.67 °C, showing an increase in suitability with higher temperatures within the threshold range of 20–30 °C, which is consistent with the results of the meta-analysis; the mean temperature of the driest quarter is 16.39 °C, the precipitation of the warmest quarter ranges from 674 to 1019 mm, the precipitation of the wettest month ranges from 262 to 319 mm, isothermality ranges from 31.40 to 33.94 °C, precipitation of the driest month ranges from 30 to 43 mm, and precipitation seasonality (coefficient of variation) ranges from 65 to 77 mm. Furthermore, when the distribution probability of *P. solenopsis* reaches its maximum, the corresponding environmental conditions include a mean temperature of the wettest quarter of approximately 28.16 °C, precipitation of the warmest quarter of approximately 774 mm, mean temperature of the driest quarter of approximately 16.39 °C, and precipitation of the wettest month of approximately 283 mm.

#### 3.2.3. Current Potential Distribution

As shown in Figure 6, the distribution map of suitable growth areas for *P. solenopsis* in China at present. The results showed that the total suitable area was about 425.67 × 10^4^ km^2^, accounting for 44.29% of China’s total land area. The high, medium, and low suitability areas accounted for 1.75%, 16.48%, and 26.06% of the total land area, respectively (Table 4). The suitable areas for the growth of *P. solenopsis* are mainly distributed in the south of the north–south boundary of China, and a few areas in Henan and Shandong provinces, of which the areas of high, medium, and low fitness areas are about 16.77 × 10^4^ km^2^ respectively.

#### 3.2.4. Potentially Suitable Climatic Distributions in the Future

Compared with the potential distribution in the current high-growth areas of China, the distribution area of *P. solenopsis* in three future emission scenarios will increase by 12.67% to 39% (Table 4). Under the SSPs1–2.6 system, by 2050, the suitable area for *P. Tinsley* will be 499.35 × 10^4^ km^2^, an increase of 17.31% compared to the current distribution area; the highly suitable area will increase by 39.07%, and the moderately suitable area will increase by 18.4%. The low suitable area will increase by 15.16%. By 2070, the total area of *P. Tinsley* will be 479.63 × 10^4^ km^2^, an increase of 12.67% compared to the current distribution area; the highly, moderately, and low suitable habitat areas will increase by 28.59%, 4.79%, and 16.59%, respectively. Under the SSPs3–7.0 system, by 2050, the suitable area for *P. Tinsley* will be 491.23 × 10^4^ km^2^, an increase of 15.4% compared to the current distribution area; the highly suitable area will increase by 40.52%, and the moderately suitable area will increase by 5.6%. The low suitable area will increase by 19.92%. By the 2070s, the total area of *P. solenopsis* will be 591.70 × 10^4^ km^2^, an increase of 39% compared to the current distribution area; the highly suitable habitat area will decrease by 2.68%, while the moderately and low suitable habitat areas will increase by 18.71% and 54.63%, respectively. Under the SSPs5–8.5 system, by 2050, the suitable area for *P. Tinsley* will be 578.13 × 10^4^ km^2^, an increase of 35.82% compared to the current distribution area; the highly suitable area will increase by 39.18%, and the moderately suitable area will increase by 16.29%. The low suitable area will increase by 47.94%. By the 2070s, the total area of *P. solenopsis* will be 539.37 × 10^4^ km^2^, an increase of 26.27% compared to the current distribution area; the highly, moderately, and low suitable habitat areas will increase by 74.40%, 14.66%, and 31.14%, respectively (Figure 7).

## 4. Discussion

Under the conditions of global climate warming, the distribution range of *P. solenopsis* is expected to increase significantly. To address the question of whether *P. solenopsis* will continue to expand its range of activity in a warmer world, we collected data from seven experimental studies on its response to temperature changes and performed a meta-analysis to evaluate these responses. The results indicate that within the temperature range of 18–35 °C, *P. solenopsis* experiences varying degrees of impact on its development cycle, with all impacts leading to a shortened development cycle. This suggests that in higher temperature conditions, the lifecycle of *P. solenopsis* may be completed more rapidly, potentially facilitating its range expansion. Using the MaxEnt model, our predictions show that under future climate warming scenarios, *P. solenopsis* will exhibit varying degrees of outward expansion within the suitable temperature range, particularly displaying a trend of expansion from south to north. This indicates that as temperatures rise, *P. solenopsis* may invade new areas, especially in northern regions. This prediction aligns with previous research results, further validating the reliability of the MaxEnt model in predicting species distribution. These findings underscore the potential impacts of global climate warming on the distribution patterns of *P. solenopsis*. As climate warming continues, its suitable habitat is expected to expand, potentially reaching new ecological areas and posing new challenges. By combining meta-analysis with MaxEnt modeling, we can more accurately predict these trends and provide scientific evidence for future monitoring and management strategies. This approach not only helps us understand how species adapt to climate change but also informs effective control measures to address the ecological and economic impacts that *P. solenopsis* may bring.

Due to the potential for significant damage and economic losses caused by *P. solenopsis* in agricultural planting areas, monitoring its spread is particularly important [35]. The spread of this pest not only poses a broad threat to crop health but also represents a serious risk to agricultural production. Therefore, timely warnings and effective management measures are crucial. Compared to other models such as BIOCLIM, DOMAIN, and GARP, the MaxEnt model can effectively predict species distribution even with limited data [36,37,38,39]. The MaxEnt model demonstrates high accuracy and reliability in predicting species distribution, particularly when data is incomplete or sample sizes are small. This model not only accurately delineates suitable habitats for the species but also provides in-depth insights into species behavior and distribution patterns [40]. In this study, the potential suitable habitats of *P. solenopsis* were simulated using the MaxEnt model. The analysis results indicated that Guangxi and Hunan are identified as the primary highly suitable regions for this species (Figure 8), which is consistent with previous research findings [41].This result further confirms the reliability and practicality of the MaxEnt model in predicting species distribution. The model’s high accuracy is crucial for future monitoring and control measures, as it provides a scientific basis for developing management strategies against *P. solenopsis* and helps predict its potential expansion under different environmental conditions. In summary, the MaxEnt model serves as a powerful tool for understanding and addressing the spread of *P. solenopsis* and is of significant value for protecting agricultural ecosystems.

It is well known that insects are poikilothermic animals, and their population distribution is significantly influenced by temperature [42,43]. Consequently, ongoing global warming has a profound impact on the distribution patterns of many insect species, leading to range expansions and changes [44,45]. In this context, temperature may act as the primary driver for the survival, development, and reproduction of *P. solenopsis*. Within suitable temperature ranges, the growth and development of *P. solenopsis* are notably promoted, while temperature variations can profoundly affect its lifecycle. In this study, we analyzed the relationship between the probability distribution of *P. solenopsis* and the dominant environmental variables and obtained the corresponding response curves. The results show that Bio18 (temperature seasonality) and Bio8 (temperature range) are the main factors influencing the distribution of *P. solenopsis*, revealing the significant role of temperature changes in determining its distribution range. Additionally, Bio3 (mean annual temperature) indicates that *P. solenopsis* slightly exceeds the predicted threshold. This may be due to the higher complexity of the field environment compared to experimental conditions. This detailed analysis of environmental variables not only aids in understanding the growth and expansion patterns of *P. solenopsis* under varying temperature conditions but also provides valuable insights for future ecological models. It will help in developing more precise control strategies and better addressing the challenges posed by climate-induced changes in insect distributions. By considering these environmental factors, we can more comprehensively predict the potential distribution changes of *P. solenopsis* and provide scientific guidance for practical management efforts.

Global warming is expected to significantly alter the functioning and structure of ecosystems, leading to changes in the distribution of biological habitats [46]. Research indicates that under three different Shared Socioeconomic Pathways (SSP) scenarios, the distribution range of *P. solenopsis* in China is projected to increase. This trend reflects a rise in the frequency and extent of insect infestations and damage, which could have profound effects on agricultural production [47]. As *P. solenopsis* continues to spread, its impact on crop yields and agricultural management may intensify, highlighting the need for proactive control measures. In this study, it is emphasized that under future climate change conditions, *P. solenopsis* is expected to migrate from low-latitude regions to higher-latitude areas. This potential range expansion may lead to the pest invading previously unaffected regions, especially as temperatures continue to rise. Understanding this potential expansion is crucial for developing effective management and control strategies. Given the importance of early warning systems for predicting insect invasion directions and ranges, the minimum distribution model established in this study provides essential information for future monitoring and warning efforts. This model will help in anticipating the potential spread of *P. solenopsis* and enable the implementation of targeted interventions to mitigate its impact. Integrating this predictive model into monitoring frameworks will better prepare us to address the challenges posed by climate-induced changes in insect distributions and develop effective protective measures.

## 5. Conclusions

In recent decades, the distribution range of *P. solenopsis* has expanded, but more effective management methods are currently lacking. It is crucial to note that the timing of chemical control is essential. Therefore, early prediction of distribution areas and preparation of chemical supplies will achieve better preventive effects. This study integrated data from 221 warming experiments and used the MaxEnt model to predict the suitable temperature for the survival of *P. solenopsis* and its current and future suitable habitats. The results showed that: (1) Within the temperature range of 20–35 °C, the developmental cycle of *P. solenopsis* shortened with increasing temperature. (2) Under three future climate scenarios, the bioclimatic variables Bio18, Bio8, and Bio3 indicated an increasing trend for *P. solenopsis*. This study provides policymakers with information to better formulate appropriate pest management strategies to prevent widespread economic losses caused by pests due to future climate warming.

## Figures and Tables

**Figure 1 insects-15-00675-f001:**
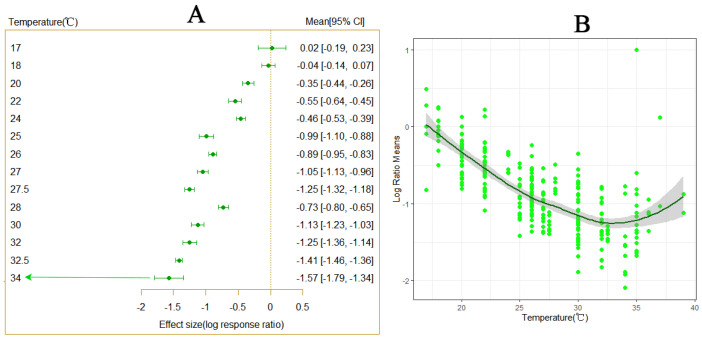
The impact of temperature variation on the developmental period of *P. solenopsis*. The (**A**) panel demonstrates a significant reduction in the developmental period of *P. solenopsis* compared to 18 °C across different temperature scales (*Q_m_* = 209.3343, *df* = 15, *p* < 0.0001). The (**B**) panel illustrates the variation in effect size under different temperature conditions. A smaller effect size indicates a greater impact on the developmental period of *P. solenopsis*.

**Figure 2 insects-15-00675-f002:**
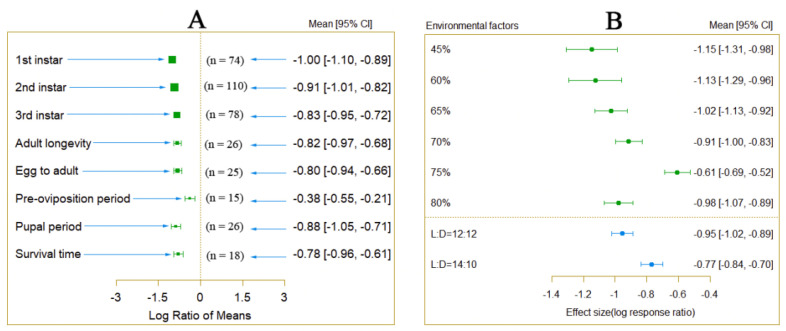
Difference in the influence of different humidity and photoperiod on the development cycle of *P. Tinsley* (*Q_m_* = 18.8807, *df* = 5, *p* < 0.0001).

**Figure 3 insects-15-00675-f003:**
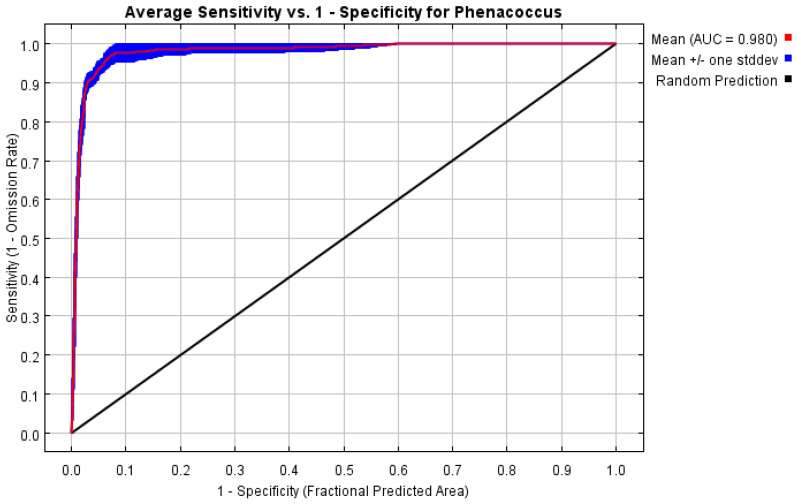
Reliability test of the distribution model created for *P. solenopsis*.

**Figure 4 insects-15-00675-f004:**
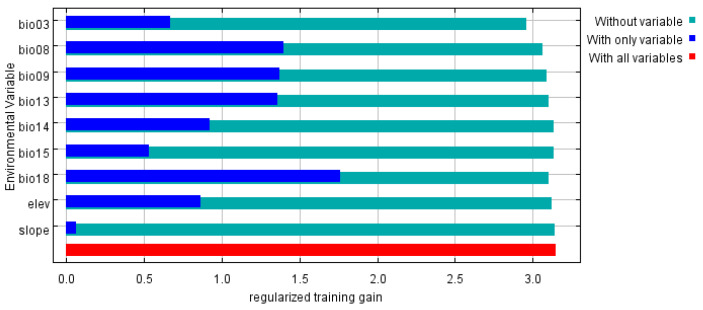
Jackknife test result of environmental factors for *P. solenopsis*.

**Figure 5 insects-15-00675-f005:**
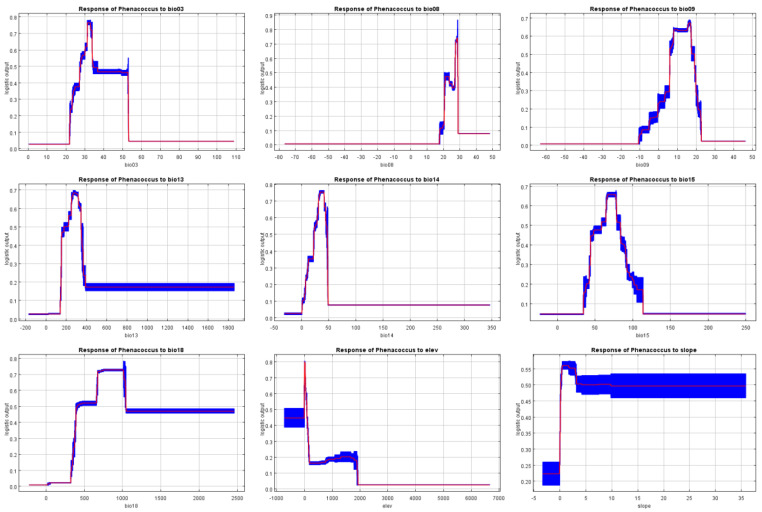
Response curves of the probability of presence for *P. solenopsis*. The red line is the average response of the MaxEnt run. The blue part is the average +/− one standard deviation.

**Figure 6 insects-15-00675-f006:**
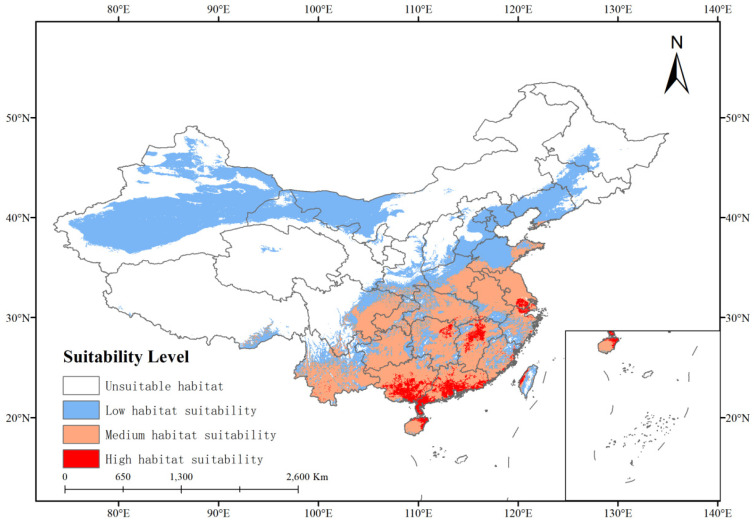
Potential current status and suitable habitats for *P. solenopsis* in China.

**Figure 7 insects-15-00675-f007:**
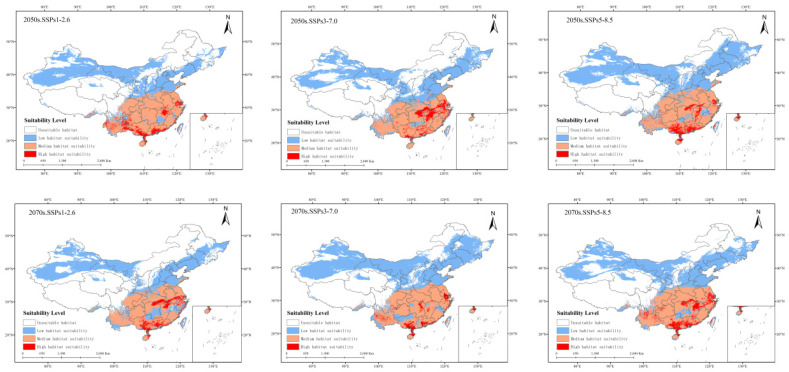
Potentially suitable climatic distribution of *P. solenopsis* under different climate change scenarios in China.

**Figure 8 insects-15-00675-f008:**
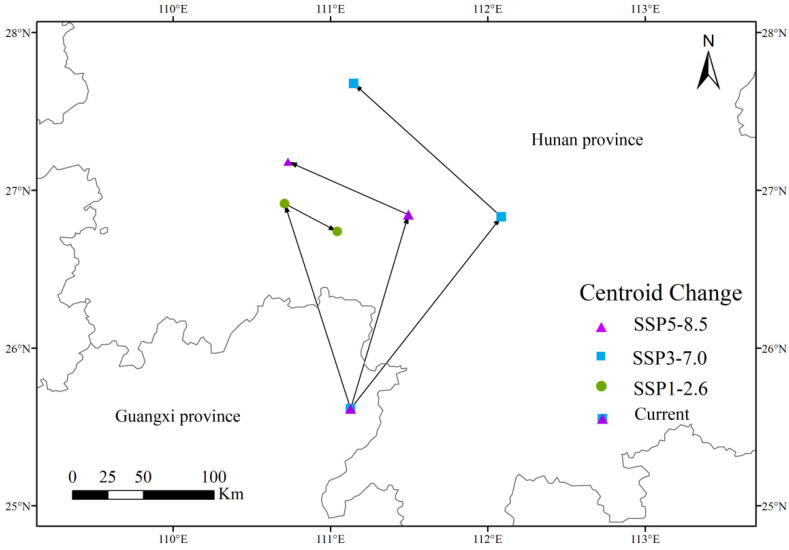
Highly suitable area centroid distributional shifts under climate change for *P. solenopsis*.

**Table 1 insects-15-00675-t001:** Three future emission scenarios.

Emission	Description
SSP1–2.6	SSP1 (Low forcing scenario) upgrade to RCP2.6 scenario (Radiative forcing reaches 2.6 W/m^2^ in 2100)
SSP3–7.0	SSP3 (Medium forcing scenario) new RCP7.0 emission path (Radiative forcing reaches 7.0 W/m^2^ in 2100)
SSP5–8.5	SSP5 (High forcing scenario) upgrade to RCP8.5 scenario (SSP5 is the only SSP scenario that can achieve radiative forcing of 8.5 W/m^2^ in 2100)

**Table 2 insects-15-00675-t002:** Environmental data used in the study.

Variable	Description	Whether to Retain after Filtering
Bio01	Annual Mean Temperature (°C)	No
Bio02	Mean Diurnal Range (Mean of Monthly (Max Temp–Min Temp)) (°C)	No
Bio03	Isothermality (BIO2/BIO7) (×100)	Yes
Bio04	Temperature Seasonality (Standard Deviation × 100)	No
Bio05	Max Temperature of Warmest Month (°C)	No
Bio06	Min Temperature of Coldest Month (°C)	No
Bio07	Temperature Annual Range (BIO5–BIO6) (°C)	No
Bio08	Mean Temperature of Wettest Quarter (°C)	Yes
Bio09	Mean Temperature of Driest Quarter (°C)	Yes
Bio10	Mean Temperature of Warmest Quarter (°C)	No
Bio11	Mean Temperature of Coldest Quarter (°C)	No
Bio12	Annual Precipitation (mm)	No
Bio13	Precipitation of Wettest Month (mm)	Yes
Bio14	Precipitation of Driest Month (mm)	Yes
Bio15	Precipitation Seasonality (Coefficient of Variation)	Yes
Bio16	Precipitation of Wettest Quarter (mm)	No
Bio17	Precipitation of Driest Quarter (mm)	No
Bio18	Precipitation of Warmest Quarter (mm)	Yes
Bio19	Precipitation of Coldest Quarter (mm)	No
Altitude	Altitude (m)	Yes
Slope	Incline (°)	Yes

**Table 3 insects-15-00675-t003:** Permutation importance of variables for modeling.

Variable	Percent Contribution (%)	Permutation Importance (%)
Bio18	40.6	5.3
Bio8	21	47.5
Bio3	17.9	29.5
Bio13	5.5	1.6
Altitude	4.5	2.3
Bio14	3.7	0.4
Bio9	3.6	11.9
Bio15	2.8	1.4
Slope	0.3	0.1

**Table 4 insects-15-00675-t004:** Suitable areas for *P. solenopsis* under different climate change scenarios (×10^4^ km^2^) and area change (%).

Period	Highly Suitable	Moderately Suitable	Low Suitable	Total Suitable
Current	16.77	158.41	250.49	425.67
2050s, SSPs1–2.6	23.33 (+39.07%)	187.55 (+18.40%)	288.47 (+15.16%)	499.35 (+17.31%)
2050s, SSPs3–7.0	23.57 (+40.52%)	167.27 (+5.60%)	300.39 (+19.92%)	491.23 (+15.40%)
2050s, SSPs5–8.5	23.35 (+39.18%)	184.21 (+16.29%)	370.58 (+47.94%)	578.13 (+35.82%)
2070s, SSPs1–2.6	21.57 (+28.59%)	166.00 (+4.79%)	292.06 (+16.59%)	479.63 (+12.67%)
2070s, SSPs3–7.0	16.32 (−2.68%)	188.05 (+18.71%)	387.32 (+54.63%)	591.70 (+39.00%)
2070s, SSPs5–8.5	29.26 (+74.40%)	181.63 (+14.66%)	328.49 (+31.14%)	539.37 (+26.27%)

## Data Availability

The data supporting the results are available in a public repository at: https://doi.org/10.6084/m9.figshare.25486894.v1, accessed on 27 March 2024; GBIF.org (27 March 2024) GBIF Occurrence Download https://doi.org/10.15468/dl.vfpt9d, accessed on 27 March 2024.

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
