# Peer review of "Meta-Analysis and MaxEnt Model Prediction of the Distribution of Phenacoccus solenopsis Tinsley in China under the Context of Climate Change"

_insects, 2024, doi:10.3390/insects15090675_

Round 1

Reviewer 1 Report

Comments and Suggestions for Authors

Dear Authors,

I have read your work with great attention and interest. Below, I offer my main comments and suggestions:

Line 25: I suggest choosing keywords that differ from those used in the title. This will help improve the visibility of your work in search results and thus increase its dissemination.

Lines 9-24: The abstract is well-structured but has some issues with clarity. I recommend reviewing the grammar and sentence flow. The section should more clearly emphasize the importance of your research. For example, why is it crucial to monitor and control Phenacoccus solenopsis? What are the practical implications of the predictions made?

Lines 10-11: The sentence "in this study, it through a comprehensive analysis" seems incomplete or poorly formulated. Could you clarify what you intend to convey? I suggest rephrasing the sentence to improve clarity.

Line 34: On line 34, "P. Tinsley" is mentioned for the first time without any prior explanation or definition. It would be helpful to include an initial note explaining that this is a synonym or alternate term for Phenacoccus solenopsis.

Lines 48-49: The transition from describing the biology of Phenacoccus solenopsis to modeling its distribution is abrupt. You might consider adding a transitional sentence that explains the connection between the pest's biology and the importance of modeling its distribution.

Lines 68-69: The paragraph describes the data collection process, but the rationale for the inclusion and exclusion criteria is not provided. It would be helpful to explain, for example, why a temperature of 18°C was chosen as a control.

Lines 99-107: The section describing the verification of data heterogeneity could be made more accessible. Consider providing a more detailed explanation or example of the method used for verifying heterogeneity.

Data Analysis section: This section contains many formulas and technical descriptions that may be complex for non-expert readers. It might be helpful to include brief, simplified explanations for each formula.

Line 143: It is unclear whether the 460 distribution points represent the full range of relevant climatic conditions. Adding a note discussing the representativeness of the data could be useful.

Discussion section: The discussion is somewhat sparse in comparison to the results obtained. I suggest revisiting the entire paragraph and expanding it. The possibility of further research or improvements in the models used is not discussed. This could be an area to explore to enhance the discussion.

Overall, the entire manuscript could benefit from a stylistic review to make the text more fluid and less technical where possible, especially in sections aimed at a broader audience.

The bibliography does not adhere to the style required by the journal and contains some errors. I encourage you to review it thoroughly.

These comments are intended to strengthen the manuscript by improving the clarity and coherence of the work presented, and I hope they are taken into consideration.

Best regards,

Comments on the Quality of English Language

The manuscript requires moderate editing for English language. While the content is strong, improvements in grammar, sentence structure, and overall readability are necessary to enhance clarity and ensure the manuscript meets the journal's standards. A thorough review of the language will help in effectively conveying the research findings to a broader audience.

Reviewer 2 Report

Comments and Suggestions for Authors

I think, that the manuscript is interesting, but I would like to suggest some changes.

In my opinion, the way the species name is written is incorrect. It should be Phenacoccus solenopsis Tinsley not P. Tinseley. The shorten version of species name should be: P. solenopsis. This should be corrected in the whole text.

Please add author’s name to all species name in the text.

A serious mistake is the lack of species affiliation. Phenacoccus solenopsis belongs to the family Pseudococcidae, superfamily Coccoidea, suborder Sternorrhyncha and order Hemiptera. Information about species systematic placement should be added.

I think, that the sentence is not clear: The Phenacoccus solenopsis poses a serious threat to global crops, and controlling climate change has become a crucial strategy. The first part and the second part of the sentence are not related to each other.

This sentence is incorrect: Its secretions can induce sooty mold, which hinders photosynthesis and may cause leaf detachment, and in severe cases, plant death. The honeydew is not secretion, but it is excreted by e.g. scale insects, so it is a waste product.

Information about distribution of Phenacoccus solenopsis should be added.

The chapter “References” was prepared carelessly. Authors’ names and titles should be written according to one pattern. This chapter needs to be improved.

Round 2

Reviewer 1 Report

Comments and Suggestions for Authors

Dear Authors,

I appreciate the changes you have made to your manuscript, which has improved significantly.

Congratulations!